

# Schrödinger approach to Mean Field Games with negative coordination

**Thibault Bonnemain[1,2,3], Thierry Gobron[2,4] and Denis Ullmo[1⋆]**

**1** Université Paris-Saclay, CNRS, LPTMS, 91405, Orsay, France.
**2** LPTM, CNRS UMR 8089, Univ. Cergy-Pontoise, 95302 Cergy-Pontoise, France.
**3** Department of Mathematics, Physics and Electrical Engineering,
Northumbria University, Newcastle upon Tyne, United Kingdom
**4** CNRS UMR 8524, Université de Lille, Laboratoire Paul Painlevé,
59655 Villeneuve d'Ascq, France

⋆ denis.ullmo@universite-paris-saclay.fr

## Abstract

Mean Field Games provide a powerful framework to analyze the dynamics of a large number of controlled agents in interaction. Here we consider such systems when the interactions between agents result in a negative coordination and analyze the behavior of the associated system of coupled PDEs using the now well established correspondence with the non linear Schrödinger equation. We focus on the long optimization time limit and on configurations such that the game we consider goes through different regimes in which the relative importance of disorder, interactions between agents and external potential vary, which makes possible to get insights on the role of the forward-backward structure of the Mean Field Game equations in relation with the way these various regimes are connected.

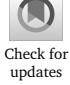

# 1 Introduction

Mean Field Games are a powerful framework introduced a little more than ten years ago by Lasry and Lions [1–3] to deal with complex problems of game theory when the number of "players" becomes large. Their applications are numerous, ranging from finance [4–6] to sociology [7–9] and engineering science [10–12], and more generally when tackling optimization issues involving many coupled subsystems.

Important mathematical efforts and progresses in this field have been made recently, for one part on the coherence of the theory [13, 14], with important results on the existence and uniqueness of a solution to these problems [15–17], and in the study of the convergence of a many player game to its mean field counterpart [18–20], and on the other part on the development of effective numerical schemes [21–24] granting the opportunity for more application oriented studies, and, especially in the more recent years, in the extension of the theory to more complex framework [5, 17, 25, 26].

However, constitutive equations of Mean Field Games are difficult to analyze. Few exact solutions exist, mainly in simplified settings [27–30], and the numerical schemes, while quan-

titatively accurate, do not provide a complete elucidation of the underlying mechanisms. This lack of general understanding on the behaviour of Mean Field Games is most presumably slowing down their appropriation by researchers concerned primarily by applications to sociology, economy, or engineering sciences.

It appears therefore useful to study a small set of paradigmatic Mean Field Game problems, which, in the spirit of the Ising problem of Statistical Mechanics, are simple enough to be fully analyzed, and "understood" – in the sense a physicist would give to that word – but complex and rich enough to shed some light on the behaviour of a much larger class of Mean Field Games. Quadratic Mean Field Games, for which the connection to non-linear Schrödinger (NLS) equation can be used to make a link with a field very familiar to physicists, are a good candidate for that role, and have been previously studied by some of us in the regime of strong positive coordination [31, 32].

In this paper, we extend the previous studies cited above to the strong negative coordination regime, when the behavior of the agents results in a repulsive interaction between them, and that this repulsion essentially dominates the dynamics (see below for a more specific statement). This will allow us in particular to address one of the conceptual difficulties posed by Mean Field Games, namely the one associated with the forward-backward structure of the equations, which poses new challenges with respect to time-forward systems of equations usually met in physics. In particular, as the system configuration at any given time depends on both initial and final conditions, conserved quantities, whenever they exist, cannot be determined a-priori but only as a by-product of the resolution of the equations for the dynamics.

To stress the specific role of the forward-backward structure, we shall moreover focus on the long optimization time limit, and choose a setting (typically a very narrow initial distribution of agents) such that the system we consider goes through different regimes in which the relative importance of disorder, interactions between agents and external potential varies, evidencing the role of the forward-backward structure in the way they are linked together.

The structure of this paper is the following: In section 2, we review briefly the Mean Field Game formalism and its connection with the non-linear Schrödinger equation and introduce a related "hydrodynamic" representation; we also address the question of conserved quantities. In section 3 we consider the *ergodic state* which, whenever it exists, is a time independent solution playing a fundamental role in the long optimization time limit we consider here. Indeed, this ergodic state not only describes a significant part of the agents dynamics, but its existence also provides a major simplification, even for the transient dynamics, as it essentially decouples the final and initial boundary conditions of the problem. In section 4 we study in details two important limiting regimes. Finally, in section 5, we consider the full dynamics of the problem, and address the important question of matching the different regimes. Section 6 contains a summary of our results and concluding remarks.

## 2 Quadratic Mean Field Games with negative coordination

### 2.1 Derivation of Mean Field Game equations

We consider a large set of players, or agents, which are described by a *state variable* $\mathbf{X}^i \in \mathbb{R}^d$, $i \in \{1, \cdots, N\}$ representing what is supposed to be their relevant characteristics in the problem at hand (physical position, amount of a given resource, social status, etc..). Those $N$ players are assumed to be identical in all their characteristics, except possibly in the initial conditions and the stochastic realizations of the dynamics.

In the simplest case, these state variables follow a Langevin dynamics

$$d\mathbf{X}_t^i = \mathbf{a}_t^i dt + \sigma d\mathbf{W}_t^i \,, \tag{1}$$

where the drift velocity $\mathbf{a}_t^i$ is the *control parameter* fixed by the agent according to his own strategy, $\sigma$ is a constant and each of the $d$ component of $\mathbf{W}^i$ is an independent white noise of variance 1. For each agent, the strategy consists in adapting his velocity in order to minimize a cost functional that reflects his preferences, averaged over all possible future trajectories

$$c[\mathbf{a}^i](t, \mathbf{x}_t^i) = \langle \int_t^T \left( L(\mathbf{X}_\tau^i, \mathbf{a}_\tau^i) - V[m_\tau](\mathbf{X}_\tau^i) \right) d\tau \rangle_{\text{noise}} + \langle c_T(\mathbf{X}_T^i) \rangle_{\text{noise}} \,. \tag{2}$$

In this expression, $\langle \cdot \rangle_{\text{noise}}$ means an average over all realisations of the noise for trajectories starting at $\mathbf{x}_t^i$ at time $t$, $L(\mathbf{x}, \mathbf{a})$ is a "running cost" depending on both state and control, and $c_T(\mathbf{x})$ is the "final cost" depending on the state of the agent at the end of the optimization period $T$. The interaction with the others players at time $t$ is given through the dependence on the empirical density of agents $m_t$ in the state space,

$$m_t(\mathbf{x}) = \frac{1}{N} \sum_i \delta(\mathbf{x} - \mathbf{X}^i(t)) \,, \tag{3}$$

which embodies the fact that the interaction is "exchangeable", namely that it depends on the positions of the players in the state space and not on their identity. The game is "quadratic" in the sense that the running cost depends quadratically on the control parameter, namely $L(\mathbf{x}, \mathbf{a}) = \mu a^2/2$. Hereafter we consider potentials which are linear functionals of the density $V[m](\mathbf{x}) = g\, m(t, \mathbf{x}) + U_0(\mathbf{x})$, where $g$ represents the strength of the interactions and $U_0(\mathbf{x})$ is a (reversed) potential defining a landscape in the state space accounting for, for instance, the proximity to various facilities or resources, trending markets, etc$\cdots$ We stress that with our sign convention, $V[m](\mathbf{x})$ has to be understood as a gain (not a cost), and thus negative values for $g$ imply repulsive interactions, and the reversed potential $U_0(\mathbf{x})$ needs to have large and negative values at large distances to be "confining".

In the limit of a very large number of players, one can assume that the density $m(t, \mathbf{x})$ becomes a deterministic object which cannot be modified by the behavior of a single player. Therefore the optimization problem (2) decouples for each player and can be solved introducing the value function $u(x, t) = \min_a c[a](t, \mathbf{x})$. Using linear programming [33], this function can be shown to evolve according to Hamilton-Jacobi-Bellman equation [2]. In turn, consistency imposes that the (yet unknown) time dependent density $m_t$ is solution of the Fokker-Planck equation associated with the (single particle) Langevin equation (1) with the velocities that realize $u(x, t)$. As a consequence, the study of Mean Field Games reduces to that of a system of two coupled PDEs [1, 2, 22, 32]

$$\begin{cases} \partial_t u(t, \mathbf{x}) = \dfrac{1}{2\mu} [\nabla u(t, \mathbf{x})]^2 - \dfrac{\sigma^2}{2} \Delta u(t, x) + g m(t, \mathbf{x}) + U_0(\mathbf{x}) & \text{[HJB]} \\[2mm] \partial_t m(t, \mathbf{x}) = \dfrac{1}{\mu} \nabla [m(t, \mathbf{x}) \nabla u(t, \mathbf{x})] + \dfrac{\sigma^2}{2} \Delta m(t, \mathbf{x}) & \text{[FP]} \end{cases} \tag{4}$$

This system of equation has a rather atypical "Forward-Backward" structure, which shows up in particular through the signs in front of the Laplacian terms, which are different in both equations. The boundary conditions also reflect this structure, as the final value of the value function is fixed by the terminal cost, $u(T, \mathbf{x}) = c_T(\mathbf{x})$, while the density of players evolves from a fixed initial distribution. Understanding the consequences of such a structure is one of the main challenges here and this paper gives a contribution in that direction through a discussion of various limiting regimes and approximation schemes.

In this respect, a key-point is the concept of ergodic state introduced in this setting by Cardaliaguet et al. [34]. In the long optimization time limit $T \to \infty$ and under some additional assumptions that are verified here, it is possible to show that for most of the duration of the game the system will stay close to a stationary state

$$
\left| \begin{aligned}
m(\mathbf{x}, t) &\simeq m_{\text{er}}(\mathbf{x}) \\
u(\mathbf{x}, t) &\simeq u_{\text{er}}(\mathbf{x}) - \lambda t
\end{aligned} \right. \qquad (\text{for } 0 \ll t \ll T) ,
\tag{5}
$$

where $m_{\text{er}}(x)$ and $u_{\text{er}}(x)$ are solutions of the time independent equations

$$
\begin{cases}
-\lambda = \dfrac{1}{2\mu} [\nabla u_{\text{er}}(\mathbf{x})]^2 - \dfrac{\sigma^2}{2} \Delta u_{\text{er}}(x) + g m_{\text{er}}(\mathbf{x}) + U_0(\mathbf{x}) \\[2mm]
0 = \dfrac{1}{\mu} \nabla [m_{\text{er}}(\mathbf{x}) \nabla u_{\text{er}}(\mathbf{x})] + \dfrac{\sigma^2}{2} \Delta m_{\text{er}}(\mathbf{x})
\end{cases} ,
\tag{6}
$$

and $\lambda$ a constant that can be determined through the normalisation of $m$.

As we shall see, this notion is instrumental to the way we look at a Mean Field Game problem. The ergodic state for the quadratic games we consider will be studied in section 3.

## 2.2 Alternative representations

Even if the forward-backward nature of Eqs. (4) constitutes the main challenge in mean field games studies, the coupling of a Fokker-Planck equation with an Hamilton-Jacobi-Bellman equation is not something physicists are particularly used to dealing with and poses its own challenges. In the special case of quadratic mean field games, however, this problem can be cast in a form more familiar to physicists [22, 32, 35]. We discuss now these alternative representations.

### 2.2.1 Schrödinger representation

Proceeding as in [32] we can define a change of variables $(u(t, \mathbf{x}), m(t, \mathbf{x})) \mapsto (\Phi(t, \mathbf{x}), \Gamma(t, \mathbf{x}))$ through the relations

$$
\begin{cases}
u(t, \mathbf{x}) = -\mu \sigma^2 \log \Phi(t, \mathbf{x}) \\
m(t, \mathbf{x}) = \Gamma(t, \mathbf{x}) \Phi(t, \mathbf{x})
\end{cases} ,
\tag{7}
$$

where the first equation is a classical Cole-Hopf transform [36] and the second corresponds to an "Hermitization" of Eq. (4). In terms of the new variables $(\Phi, \Gamma)$ the Mean Field Game equations reads

$$
\begin{cases}
-\mu \sigma^2 \partial_t \Phi = \dfrac{\mu \sigma^4}{2} \Delta \Phi + (U_0 + g\Gamma\Phi)\Phi \\[2mm]
+\mu \sigma^2 \partial_t \Gamma = \dfrac{\mu \sigma^4}{2} \Delta \Gamma + (U_0 + g\Gamma\Phi)\Gamma
\end{cases} .
\tag{8}
$$

As for the original form of the Mean Field Games equations this system has a forward-backward structure brought both by opposite relative signs for the differential terms in the two equations and by mixed initial and final boundary conditions $\Phi(T, x) = \exp[-c_T(x)/\mu\sigma^2]$, $\Gamma(0, x)\Phi(0, x) = m_0(x)$. Through these transformations the system (4) can be mapped onto the non-linear Schrödinger equation

$$
i\hbar \partial_t \Psi = -\frac{\hbar^2}{2\mu} \Delta \Psi - (U_0 + g\rho)\Psi ,
\tag{9}
$$

under the formal correspondence $\mu\sigma^2 \to \hbar$, $\Phi(\mathbf{x}, t) \to \Psi(\mathbf{x}, it)$, $\Gamma(\mathbf{x}, t) \to [\Psi(\mathbf{x}, it)]$ and $\rho \equiv \|\Psi\|^2 \to m \equiv \Phi\Gamma$. Equations (8) differ from non-linear Schrödinger in a few ways. First,

they retain the forward-backward structure inherited from the original Mean Field Game equations, and the functional space of which their solutions $\Phi$ and $\Gamma$ can be constructed also differs. Actually $\Phi$ and $\Gamma$ are non-periodic, real positive functions, while $\Psi$ would be a complex valued function. Those differences are significant but are not important enough to undermine the value of this mapping. Non-linear Schrödinger equation has been studied for decades in the various fields of non-linear optics [37], Bose-Einstein condensation [38] or fluid dynamics [39]. Several methods have been developed along the years to deal with this equation and most can be adapted to mean field games [32].

### 2.2.2 Hydrodynamic representation

Starting from the non-linear Schrödinger representation of Eqs. (4) it is also possible to exploit the "Hermitized" nature of the previous transformations and perform a Madelung-like transformation [40]

$$\begin{cases} \Phi(t,\mathbf{x}) = \sqrt{m(t,\mathbf{x})}e^{K(t,\mathbf{x})} \\ \Gamma(t,\mathbf{x}) = \sqrt{m(t,\mathbf{x})}e^{-K(t,\mathbf{x})} \end{cases}. \tag{10}$$

Defining a velocity $\mathbf{v}$ as

$$\mathbf{v} \equiv \sigma^2 \nabla K = \sigma^2 \frac{\Gamma\nabla\Phi - \Phi\nabla\Gamma}{2m} = -\frac{\nabla u}{\mu} - \sigma^2 \frac{\nabla m}{2m}, \tag{11}$$

it is easy from equations (8) to obtain a continuity equation along with its associated Euler equation

$$\begin{cases} \partial_t m + \nabla.(mv) = 0 \\ \partial_t \mathbf{v} + \nabla\left[\dfrac{\sigma^4}{2\sqrt{m}}\Delta\sqrt{m} + \dfrac{v^2}{2} + \dfrac{gm + U_0}{\mu}\right] = 0 \end{cases}, \tag{12}$$

typical of hydrodynamics. This system closely resembles the original mean field game equations (4) but can prove to be more convenient when performing some approximations (small noise limit) or applying some specific methods of resolution.

## 2.3 Action, and conserved quantities

The system of equations (8) can be derived from stationarity of an action functional $S$ defined as

$$S[\Gamma,\Phi] \equiv \int_0^T dt \int_{\mathbb{R}} dx \left[\frac{\mu\sigma^2}{2}(\Gamma\partial_t\Phi - \Phi\partial_t\Gamma) - \frac{\mu\sigma^4}{2}\nabla\Gamma.\nabla\Phi + \left[U_0 + \frac{g}{2}\Gamma\Phi\right]\Gamma\Phi\right], \tag{13}$$

so that

$$\text{Eq. (8)} \quad \Leftrightarrow \quad \begin{cases} \dfrac{\delta S}{\delta\Phi} = 0 \\ \dfrac{\delta S}{\delta\Gamma} = 0 \end{cases}. \tag{14}$$

The existence of an action underlying the dynamics has two consequences. First, and as we shall see in section 4, this action can serve as the basis of a variational approach. Second, using Noether theorem, time translation invariance implies that there exists a related conserved quantity that, by analogy with physical systems, we shall call "energy".

Depending on the considered regime of approximation, either the Schrödinger or hydrodynamic representation may prove to be more convenient. As such, we provide the reader

with two alternative expressions for the energy of the game

$$
\begin{aligned}
E &= \int_{\mathbb{R}} dx \left[ -\frac{\mu\sigma^4}{2} \nabla\Gamma.\nabla\Phi + U_0\Gamma\Phi + \frac{g}{2}(\Gamma\Phi)^2 \right] \\
&= \int_{\mathbb{R}} dx \left[ \frac{\mu\sigma^2}{2}\left( m\left(\frac{||\mathbf{v}||}{\sigma}\right)^2 - \sigma^2\frac{(\nabla m)^2}{4m} \right) + U_0\, m + \frac{g}{2}\, m^2 \right].
\end{aligned}
\tag{15}
$$

Note however that this quantity cannot be computed from the boundary data without solving the dynamics. In analogy with physical systems, each of the three terms under the integral can be given an interpretation: the first, $\sigma$ dependent, is a "kinetic" energy, and the two others are, respectively, a "potential" energy and an interaction energy.

In the following sections, we are going to consider different regimes of approximation, which will be characterized by a different balance between the various components of the energy. The conservation of total energy, and the fact that a transition from one regime to another implies a transfer between one "kind" of energy to another, will help us providing a global picture, across the various regimes, of the Mean Field Game dynamics.

## 3 Static Mean Field Game: the ergodic state

The notion of ergodic state is crucial in Mean Field Games theory, and its importance is twofold. To start with, it corresponds to a simpler, static, problem, which, when it exists can provide a good approximation of the behaviour of solutions of Eqs. (4) for all intermediate times. It also allows for the initial and final parts of the dynamics (when entering or leaving the ergodic state) to essentially decouple. Instead of constructing a solution of Eqs. (4) associated with the pair of boundary conditions $m_0(x)$ and $c_T(x)$, the initial dynamics can be approximated by a simpler Mean Field Game, with the same arbitrary initial condition $m_0(x)$ but a generic terminal condition: the ergodic state. Conversely, the final part of the dynamics can be described by another Mean Field game, starting in the ergodic state and evolving with $c_T(x)$ as the final condition. In this way, this notion of ergodic state reduces the dynamical problem with mixed boundary conditions (4) to two relatively simpler ones. The aim of this section is thus to describe the ergodic solution, and the possible approximation schemes that can be used to describe it, as well as discuss its stability.

In the strong interaction regime we focus on, the ergodic state can be approached equivalently within the NLS representation and the hydrodynamic one. Both approaches lead to a very simple analysis, we present both below.

### 3.1 Alternative representations in the ergodic state

In the ergodic state, strategies become stationary, as established by Eqs. (5). In the Schrödinger representation, the ergodic solutions of the equations (8) depend on time through an overall scaling factor as

$$
\begin{cases}
\Phi(t,\mathbf{x}) = \exp\left(+\frac{\lambda}{\mu\sigma^2}t\right)\Phi_{\mathrm{er}}(\mathbf{x}) \\
\Gamma(t,\mathbf{x}) = \exp\left(-\frac{\lambda}{\mu\sigma^2}t\right)\Gamma_{\mathrm{er}}(\mathbf{x})
\end{cases},
\tag{16}
$$

where $\Phi_{\mathrm{er}}(\mathbf{x}) = \exp\left[-u_{\mathrm{er}}(\mathbf{x})/\mu\sigma^2\right]$ and $\Gamma_{\mathrm{er}}(\mathbf{x}) = m_{\mathrm{er}}(\mathbf{x})/\Phi_{\mathrm{er}}(\mathbf{x})$. Furthermore, boundary conditions for the density $\lim_{\mathbf{x}\to\pm\infty} m_{\mathrm{er}}(\mathbf{x}) = 0$ imply in turn that the functions $\Gamma_{\mathrm{er}}$ and $\Phi_{\mathrm{er}}$ are equal up to a multiplicative constant, so it is appropriate to relate them to $\Psi_{\mathrm{er}}(\mathbf{x})$, solution of the following stationary NLS equation

$$
-\lambda\Psi_{\mathrm{er}}(\mathbf{x}) = \frac{\mu\sigma^4}{2}\Delta\Psi_{\mathrm{er}}(\mathbf{x}) + U_0(\mathbf{x})\Psi_{\mathrm{er}}(\mathbf{x}) + g|\Psi_{\mathrm{er}}(\mathbf{x})|^2\Psi_{\mathrm{er}}(\mathbf{x}).
\tag{17}
$$

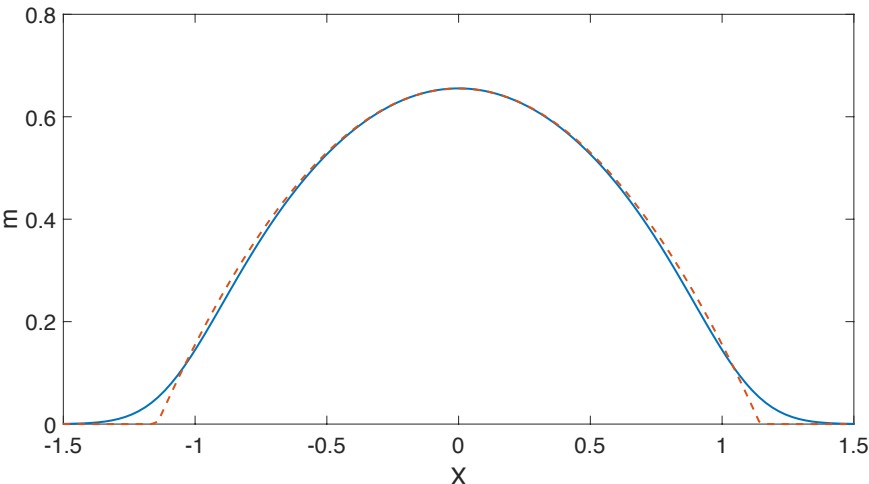

Figure 1: Computational solution of the Gross-Pitaevskii equation (full line) and Thomas-Fermi approximation (dashed line). In this case $g = -2$, $\sigma = 0.4$, $\mu = 1$ and $U_0(x) = -x^2$ $(d = 1)$.

Hence, assuming that the system is in the ergodic state, resolution of the time-dependent coupled PDEs Eqs. (8) reduces to that of the single, time-independent, ODE Eq. (17), whith $\Phi(t, \mathbf{x})\Gamma(t, \mathbf{x}) = \Phi_{\mathrm{er}}(\mathbf{x})\Gamma_{\mathrm{er}}(\mathbf{x}) = |\Psi_{\mathrm{er}}(\mathbf{x})|^2 = m_{\mathrm{er}}(\mathbf{x})$, showing in particular a direct relation between the solution $\Psi_{\mathrm{er}}$ of the stationary NLS equation (17) and the static ergodic density.

In the hydrodynamic representation, we can derive in a similar way the equations for the ergodic state. Denoting $\mathbf{v}_{\mathrm{er}}$ the ergodic velocity, Eqs. (12) readily become

$$\begin{cases} \mathbf{v}_{\mathrm{er}} = 0 \\ \lambda + \dfrac{\sigma^4}{2\sqrt{m_{\mathrm{er}}}}\Delta\sqrt{m_{\mathrm{er}}} + \dfrac{g\, m_{\mathrm{er}} + U_0}{\mu} = 0 \end{cases}. \tag{18}$$

This form shows the stationarity of the optimal strategy, and the autonomous equation obeyed by the density in the ergodic state.

## 3.2  Bulk of the distribution: Thomas-Fermi approximation

One of the many interests of the Schrödinger representation is that we can exploit the large literature surrounding this equation. In the large interaction regime, the stationary NLS (or Gross-Pitaevskii) equation can be accurately analysed through the use of Thomas-Fermi approximation [41].

First, by looking at the expression for the energy (15), one can note that a natural length scale

$$\nu \equiv \frac{\mu\sigma^4}{|g|}, \tag{19}$$

appears. Indeed denoting $L$ the length scale characterizing a solution of Eqs. (8), we find that the "kinetic energy" behaves as

$$E_{\text{kin}} = -\int_{\mathbb{R}} dx \frac{\mu \sigma^4}{2} \nabla \Gamma . \nabla \Phi \sim \frac{\mu \sigma^4}{L^2} \, , \tag{20}$$

while the "interaction energy" behaves as

$$E_{\text{int}} = \int_{\mathbb{R}} dx \frac{g}{2} (\Phi \Gamma)^2 \sim \frac{g}{L} \, . \tag{21}$$

The ratio between kinetic and interaction energies, which is a good measure of the relative importance of the diffusion and interaction processes, is then given by

$$\left| \frac{E_{\text{kin}}}{E_{\text{int}}} \right| \sim \frac{\nu}{L} \, . \tag{22}$$

In the context of the non-linear Schrödinger equation, $\nu$ is known as the "healing length", and represents the typical length-scale on which the interaction energy balances quantum pressure (or diffusion in the context of MFG), and is named in this way because it is the minimum distance from a local perturbation at which the wave function can recover its bulk value (hence "heal").

In the limiting case where the kinetic energy is negligible in the bulk of the distribution, i.e. when the typical extension of the distribution is large in front of the healing length $\nu$ (something that, we assume, will happen because - strong - repulsive interactions will cause agents to spread despite the confining potential $U_0$), Eq.(17) reduces to a simple algebraic equation

$$-\lambda \approx U_0(\mathbf{x}) + g |\Psi_{\text{er}}(\mathbf{x})|^2 \, , \tag{23}$$

which is easily solved as

$$\Psi_{\text{TF}}(\mathbf{x}) = \begin{cases} \left( \dfrac{\lambda + U_0(\mathbf{x})}{|g|} \right)^{1/2} & \text{if } \lambda > -U_0(\mathbf{x}) \\ 0 & \text{otherwise} \end{cases} \, , \tag{24}$$

where the constant $\lambda$ is then computed using the normalisation condition

$$\int_{-\infty}^{\infty} m_{\text{er}}(\mathbf{x}) d\mathbf{x} = 1 \, . \tag{25}$$

The very same approximation can also be derived by neglecting the $o(\sigma^4)$ term in Eqs. (18), which yields

$$\begin{cases} \mathbf{v}_{\text{er}} = 0 \\ m_{\text{er}}(\mathbf{x}) = \dfrac{\lambda + U_0(\mathbf{x})}{|g|} \end{cases} \, , \tag{26}$$

an expression clearly equivalent to Eq. (24).

Such an approximation may seem naive at first but actually yields rather good results. Let us take the example of quadratic external potential $U_0(x) = -\mu \omega_0^2 x^2 / 2$. [Note that, as mentioned above, $U_0(x)$ has to be understood as a gain and, to be "confining" has to reach its maximum value for a finite $x$ and go to $-\infty$ for large $x$, thus the negative sign.] We find $\lambda = \left[ 3|g| \sqrt{\mu \omega_0^2} / 4\sqrt{2} \right]^{2/3}$, and we can see on Fig. 1 that, in the bulk, the approximation agrees perfectly with the exact (numerical) result.

The tails of the distribution, for which densities is low, and thus interactions effects are small, cannot be described in this way however and call for a specific treatment.

### 3.3 Tails of the distribution: semi-classical approximation

If Thomas-Fermi approximation yields good results in the bulk of the distribution, i.e. for $\lambda > -U_0(\mathbf{x})$, it fails to describe regions where the density of agents is small. When this density is sufficiently small however, that is in the tails of the distribution where $\lambda + U_0(\mathbf{x})$ is sufficiently negative, the problem simplifies once again because the non-linear interacting term is negligible. In this context Eq. (17) reads

$$-\lambda\Psi(\mathbf{x}) \approx \frac{\mu\sigma^4}{2}\Delta\Psi(\mathbf{x}) + U_0(\mathbf{x})\Psi(\mathbf{x})\,, \tag{27}$$

and we can solve it within a semi-classical approximation. More specifically, we look for solutions of Eq. (27) in the form $\Psi_{\text{SC}}(\mathbf{x}) = \psi(\mathbf{x})\exp\left(\frac{S(\mathbf{x})}{\sqrt{\mu\sigma^4}}\right)$ up to the second order in $\sigma^2$. As an example, we will once again look at the case of one dimensional quadratic external potential $U_0(x) = -\mu\omega_0^2 x^2/2$, and compare the approximation to numerical results. In order to keep the core of the text concise, details of the computation are provided in Appendix (A). The semi-classical approximation yields

$$\Psi_{\text{SC}}(x) = \left[\frac{C}{\mu\omega_0^2 x^2 - 2\lambda}\right]^{1/4}\exp\left\{\frac{\lambda}{\mu\omega_0\sigma^2}\left[x\sqrt{\frac{\mu\omega_0^2}{2\lambda}}\sqrt{x^2\frac{\mu\omega_0^2}{2\lambda}-1}\right.\right.$$
$$\left.\left.-\operatorname{argcosh}\left(x\sqrt{\frac{\mu\omega_0^2}{2\lambda}}\right)\right]\right\}\,, \tag{28}$$

where $C$ is a constant numerically determined to match with the bulk of the distribution. This expression gives results in very good agreement with the true solution for $x \gg X$, where the "turning point" $X \equiv \sqrt{\frac{2\lambda}{\mu\omega_0^2}}$ corresponds to the position where $\Psi_{\text{TF}}$ vanishes. Eq. (28), however, exhibits a singularity at the turning point $X$. This spurious divergence can be easily avoided by a uniform approximation [42], leading to

$$\Psi_{\text{SC}} = \begin{cases} C_{\text{left}}\left(\frac{8\pi S_{\text{left}}}{3U_0}\right)^{1/2}\cos\left(\frac{\pi}{3}\right)\left[J_{1/3}(S_{\text{left}})+J_{-1/3}(S_{\text{left}})\right] & \text{if } x < X \\ 2C_{\text{right}}\left(\frac{8S_{\text{right}}}{\pi\,|U_0|}\right)^{1/2}\cos\left(\frac{\pi}{3}\right)K_{1/3}(S_{\text{right}}) & \text{if } x > X \end{cases}\,, \tag{29}$$

where $C_{\text{left}}$ and $C_{\text{right}}$ are constants to be numerically determined, $J_\gamma$ stands for the Bessel function of the first kind of order $\gamma$ and $K_\gamma$ for the modified Bessel function of the second kind. Explicit expressions for the actions $S_{\text{left}}(x)$ and $S_{\text{right}}(x)$, in the case of the quadratic gain $U_0(x) = -\mu\omega_0^2 x^2/2$, are provided in Appendix A. Figure 2 illustrates how this uniform approximation Eq. (29) constitutes a neat improvement over Eq. (24) when describing the tails of the distribution.

Depending on the external potential $U_0(x)$, computing this approximation may become somewhat involved. If so, the tails of the distribution can still be described by an Airy function, as discussed in [41], using the consistently simpler, albeit less accurate, approximation method of linearizing the potential around $x \approx X$ and looking at the asymptotic behaviour.

### 3.4 Some properties of the ergodic state

To conclude this section on the ergodic state, we shall describe here some of its properties that will become relevant when trying to connect it to the beginning (or end) of the game.

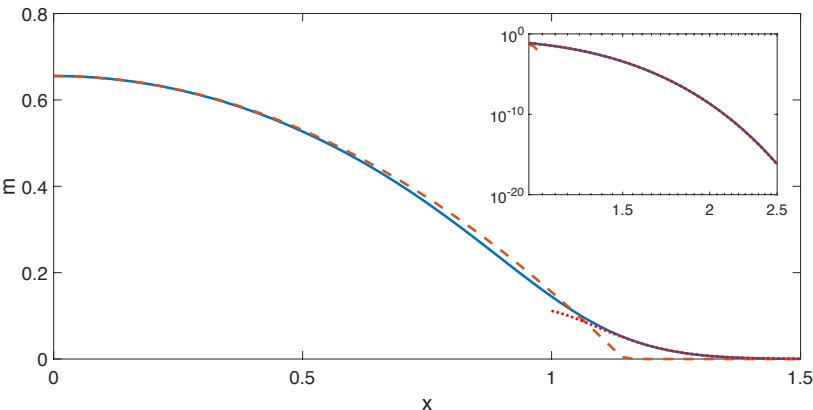

Figure 2: Computational solution of the Gross-Pitaevskii equation (full), Thomas-Fermi approximation (dashed) and semi-classical uniform approximation (dot). The inset shows the same curves in Log-Linear plot focusing on the tail of the distribution. Parameters for this figure are $g = -2$, $\sigma = 0.4$, $\mu = 1$, $U_0(x) = -x^2$ and $C = 8.10^{-4}$ ($d = 1$).

### 3.4.1 Final cost and energy in the ergodic state

Something that may not appear clearly from the definition Eqs. (6) of the ergodic state, but becomes obvious when looking at its hydrodynamic counterpart Eqs. (26), is that for quadratic Mean Field Games in the strong repulsive interaction regime, the value function $u$, becomes essentially flat during the ergodic state

$$\mathbf{v}_{\text{er}} = 0 \quad \Leftrightarrow \quad u_{\text{er}} = K_{\text{er}} + o(\sigma^2), \tag{30}$$

where the $O(\sigma^2)$ terms are the first order corrections to the Thomas Fermi approximation and $K_{\text{er}}$ is a constant. The Mean Field Games equations Eqs. (4) being invariant by translation of $u$, we will choose this constant $K_{\text{er}}$ to be zero for the rest of this paper. This characterization, $u_{\text{er}} = 0$, will then be used as an "effective" terminal condition when discussing the beginning of the game.

Another interesting aspect of the ergodic state is that it provides us with a way of computing the (conserved) energy $E = E_{\text{er}}$ of the system

$$E_{\text{er}} = \int_{\mathbb{R}^d} d\mathbf{x} \left[ \frac{g}{2} m_{\text{er}}^2 + m_{\text{er}} U_0 \right] < 0, \tag{31}$$

which is the correct expression for the "kinetic" energy up to $o(\sigma^4)$ terms. Since interactions are assumed repulsive and the external potential confining (which implies it can be chosen negative for all $\mathbf{x}$), both terms in the energy have to be negative.

With those two properties at hands, we can restrict our analysis of the transient states to games with negative energy and zero terminal conditions, making for a simpler discussion of the time-dependent problem.

### 3.4.2 Approaching the ergodic state: stability analysis

To finish this section, we discuss the stability of the ergodic state. Focusing on the bulk of the distribution we will use the hydrodynamic representation as it is the better framework to deal with the small $\sigma$ limit. Recalling Eqs. (26), the expression of the ergodic state under this representation

$$
\begin{cases}
\mathbf{v}_{\mathrm{er}} = 0 \\
m_{\mathrm{er}}(\mathbf{x}) = -\dfrac{\lambda + U_0(\mathbf{x})}{g}
\end{cases}, \tag{32}
$$

we then apply small perturbations $\delta m$ and $\delta v$ to this stationary state and compute their evolution. Near the ergodic state Eqs. (12) become

$$
\begin{cases}
\partial_t(\delta m(\mathbf{x}, t)) = -\nabla(m_{\mathrm{er}}(\mathbf{x})\delta\mathbf{v}(\mathbf{x}, t)) \\
\partial_t(\delta\mathbf{v}(\mathbf{x}, t)) = -\dfrac{g}{\mu}\nabla\delta(m(\mathbf{x}, t))
\end{cases}, \tag{33}
$$

implying

$$
\partial_{tt}(\delta m(\mathbf{x}, t)) = \frac{g}{\mu}\nabla(m_{\mathrm{er}}(\mathbf{x})\nabla\delta m(\mathbf{x}, t)). \tag{34}
$$

Assuming that $\delta m = \delta m_0 e^{\omega t}$, Eq. (34) amounts to the eigenvalue problem $\hat{D}\delta m_0 = -(\mu/g)\omega^2\delta m_0$ with

$$
\hat{D} \equiv -\nabla(m_{\mathrm{er}}(\mathbf{x})\nabla\cdot). \tag{35}
$$

It is relatively straightforward to show that $\hat{D}$ is a real symmetric operator, implying its eigenvalues are real, and furthermore that all these eigenvalues are positive (cf Appendix. B). Noting $(\epsilon_i)_{i\geq 0}$ the set of (real, positive) eigenvalues of $\hat{D}$ and $(\varphi_i(\mathbf{x}))_{i\geq 0}$ the corresponding eigenvectors, the "linear modes" in the vicinity of $(m_{\mathrm{er}}, v_{\mathrm{er}})$ are

$$
\mathcal{Q}_{(i)}^{\pm} = (\delta m_{(i)}, \delta v_{(i)}^{\pm}) \equiv \left(\varphi_i(\mathbf{x}), \pm\sqrt{-g/\mu\,\epsilon_i}\nabla\varphi_i(\mathbf{x})\right), \tag{36}
$$

and they follow an exponential time dependence $\mathcal{Q}_{(i)}^{\pm}(t) = e^{\pm\omega_i t}\mathcal{Q}_{(i)}^{\pm}(0)$, with $\omega_i = \sqrt{-g\epsilon_i/\mu}$ (remember $g < 0$).

This exponential behaviour highlights the fact that, as discussed in [32] in a simpler (variational) context, the ergodic state should be understood as a *unstable / hyperbolic* fixed point, which is approached exponentially fast at small times, and left exponentially quickly near $T$.

Returning to the particular case of the 1d quadratic external potential $U_0 = -\mu\omega_0^2 x^2/2$, and assuming as above that $\delta m \propto e^{\pm\omega t}$, we get

$$
\begin{cases}
-2\left(\dfrac{\omega}{\omega_0}\right)^2 \delta m = \partial_y\left[(1 - y^2)\partial_y\delta m\right] \\
y = x\sqrt{\dfrac{\mu\omega_0^2}{2\lambda}}
\end{cases}, \tag{37}
$$

a Legendre equation defined for $0 \leq y \leq 1$. Dismissing odd ones, the solution with smallest eigenvalue (for $\omega = \omega_0$) is the first order Legendre polynomial of the second kind, hence

$$
\delta m \approx Q_1(y)e^{\pm\omega_0 t}. \tag{38}
$$

The effect of this perturbation is thus simply to add tails to the distribution of agents.

# 4 Time dependent problem: the beginning of the game

For the sake of simplicity, we specify from now on to the one dimensional case $d = 1$ (although most of the analysis below can be extended easily to higher dimensionality). As shown by Eqs. (19)-(20)-(21)-(22), different length scales are associated with different dynamical regimes: very short distances $L \ll \nu$ are dominated by diffusion, and for $L \gg \nu$ interactions take over. The "large interaction limit" that we consider here essentially means that the healing length $\nu$ is much smaller than any characteristic feature of the "one-body" gain $U_0(x)$, and we will work under that hypothesis. However, as the size of the distribution of agents further increases, interaction effects become weaker (although the effects of diffusion decrease even more rapidly) and, even in the large $|g|$ limit that we mostly consider here, the ergodic state is still characterized by a balance between the interaction energy $E_{\text{int}}$ and the potential energy $E_{\text{pot}}$. The fact that this balance has to be reached is eventually what fixes the typical size of the ergodic state distribution.

A good setting which may allow to explore all dynamical regimes is to consider an extremely narrow initial distribution (so that its width $\Sigma_0$ is significantly smaller than $\nu$). The beginning of the game will therefore mainly consist in an expansion of this initial distribution, expansion that will go on until the balance between $E_{\text{int}}$ and $E_{\text{pot}}$ is reached. During that expansion we may neglect the effects of the external potential. In this section we will therefore study the set of equations (8) in the particular case of $U_0(x) = 0$

$$\begin{cases} -\mu\sigma^2\partial_t\Phi(t,x) = \dfrac{\mu\sigma^4}{2}\partial_{xx}\Phi(t,x) + g\Phi^2(t,x)\Gamma(t,x) \\[4mm] +\mu\sigma^2\partial_t\Gamma(t,x) = \dfrac{\mu\sigma^4}{2}\partial_{xx}\Gamma(t,x) + g\Phi(t,x)\Gamma^2(t,x) \end{cases} . \tag{39}$$

While it can be shown that this system is integrable (in the sense that there exists a canonical transform from $(\Phi, \Gamma)$ to action-angle variables) [43], we will not attempt here to explicitly use this property and will approach the various limiting regimes through the use of variational ansätze. Furthermore, as we know (cf Section 3.4) that the value function of the ergodic state, which can here be interpreted as a final cost for the beginning of the game, is essentially constant, we shall work below under the assumption that the terminal cost is essentially flat.

## 4.1 Large $\nu$ regime : Gaussian Ansatz

When the extension of the distribution of agents is small in front of $\nu$, the effects of diffusion become dominant, and Eqs. (39) become simple heat equations, for which the Green's function has a Gaussian shape. It is therefore natural to tackle this regime using Gaussian variational approach [44], as already applied to Mean Field Games in [32].

### 4.1.1 Preliminary definitions

Variational approximation amounts to minimizing the action on a small subclass of functions (here taken so that the distribution of agents is Gaussian), effectively reducing a problem with an infinite number of degrees of freedom to one with a finite, easily manageable, number. As in [32] we consider the following Ansatz

$$\begin{cases} \Phi(x,t) = \exp\left[\dfrac{(-\Lambda_t/4 + P_t \cdot x)}{\mu\sigma^2}\right] \dfrac{1}{(2\pi\Sigma_t)^{1/4}} \exp\left[-\dfrac{(x-X_t)^2}{(2\Sigma_t)^2}(1 - \dfrac{\Lambda_t}{\mu\sigma^2})\right] \\[4mm] \Gamma(x,t) = \exp\left[\dfrac{(+\Lambda_t/4 - P_t \cdot x)}{\mu\sigma^2}\right] \dfrac{1}{(2\pi\Sigma_t)^{1/4}} \exp\left[-\dfrac{(x-X_t)^2}{(2\Sigma_t)^2}(1 + \dfrac{\Lambda_t}{\mu\sigma^2})\right] \end{cases} , \tag{40}$$

which indeed yields a Gaussian distribution centered in $X_t$ with standard deviation $\Sigma_t$

$$m(t,x) = \Gamma(t,x)\Phi(t,x) = \frac{1}{\sqrt{2\pi\Sigma_t^2}} \exp\left[-\frac{(x-X_t)^2}{2(\Sigma_t)^2}\right] , \tag{41}$$

and where $P_t$ and $\Lambda_t$ respectively are the momentum and the position-momentum correlator of the system. Inserting this variational ansatz in the action (13) we get $\tilde{S} = \int_0^T \tilde{L}(t)dt$ where the Lagrangian $\tilde{L} = \tilde{L}_\tau + \tilde{E}_{\text{kin}} + \tilde{E}_{\text{int}} + \tilde{E}_{\text{pot}}$ only depends on $X_t$, $P_t$, $\Sigma_t$, $\Lambda_t$ and their time derivatives. This yields

$$\begin{cases} \tilde{L}_\tau = \dot{P}_t X_t - \dfrac{\Lambda_t}{2\Sigma_t}\dot{\Sigma}_t & \tilde{E}_{\text{kin}} = \dfrac{P_t}{2\mu} + \dfrac{\Lambda_t^2 - \mu^2\sigma^4}{8\mu\Sigma_t^2} \\[2ex] \tilde{E}_{\text{int}} = \dfrac{g}{4\sqrt{\pi}\Sigma_t} & \tilde{E}_{\text{pot}} = \displaystyle\int_{\mathbb{R}} U_0(x)m(t,x)dx \end{cases} . \tag{42}$$

As long as the density of players $m(t,x)$ remains narrow enough that $U_0(x)$ can be linearized on the distance $\Sigma_t$, we see that $\tilde{E}_{\text{pot}} \approx U_0(X_t)$ and that the variable $(X_t, P_t)$ and $(\Sigma_t, \Lambda_t)$ decouple. As discussed in [32] $(X_t, P_t)$ then follows the dynamics of a point particle of mass $\mu$ subject to the external potential $U_0(x)$. The discussion below, in which we assume $U_0(x) = 0$, could also therefore be generalized straightforwardly to this situation (by just adding the motion of the center of mass).

### 4.1.2 Evolution of the reduced system $(X_t, \Sigma_t; P_t, \Lambda_t)$ for $U_0(x) = 0$

Minimizing the action with respect to each parameter yields the evolution equations

$$\begin{cases} \dot{X}_t = \dfrac{P_t}{\mu} & \dot{P}_t = 0 \\[2ex] \dot{\Sigma}_t = \dfrac{\Lambda_t}{2\mu\Sigma_t} & \dot{\Lambda}_t = \dfrac{\Lambda_t^2 - \mu^2\sigma^4}{2\mu\Sigma_t^2} + \dfrac{g}{2\sqrt{\pi}\Sigma_t} \end{cases} . \tag{43}$$

Under the assumption that $U_0(x) = 0$, $P_t$ is a constant and is essentially a measure of the asymmetry of $\Phi(t,x)$ and $\Gamma(t,x)$ as well as the drift of the center of mass of the density. If $\Phi(t,x)$ and $\Gamma(t,x)$ are symmetric with respect to $x = x_0$, $P_t = 0$ and the center of mass does not move. For the sake of simplicity, let us focus on this configuration and let $X_t = x_0 = 0$. The equations for $(\Sigma_t; \Lambda_t)$ have a first integral corresponding to conservation of total energy of the system $\tilde{E}_{\text{tot}} = \tilde{E}_{\text{kin}} + \tilde{E}_{\text{int}} + \tilde{E}_{\text{pot}}$. Its expression reduces here to

$$\tilde{E}_{\text{tot}} = \frac{\mu\dot{\Sigma}^2}{2} - \frac{\mu\sigma^4}{8\Sigma_t^2} + \frac{g}{4\sqrt{\pi}\Sigma_t} . \tag{44}$$

### 4.1.3 Zero-energy solution

In the limit where the external potential has very slow variations (e.g: $U_0(x) = f(\epsilon x)$ with $\epsilon \ll 1$ and $f$ a smooth function such that $\lim_{x\to\pm\infty} f(x) \to -\infty$ ), the potential energy $\tilde{E}_{\text{pot}}$ can be neglected for all relevant values of $x$ but still ensures the existence of an ergodic state, characterized by a low density $m_{\text{er}}(x) \approx \epsilon$, large spreading $\Sigma_{\text{er}} \approx \epsilon^{-1}$ and small energy $\tilde{E}_{\text{tot}} \approx \epsilon$. In the limit $\epsilon \to 0$, convergence to the (asymptotic) ergodic state occurs in the limit of an infinitely long game, $T \to \infty$, and is characterized by an indefinite spreading with zero total energy, $\tilde{E}_{\text{tot}} = 0$. In that case the evolution equation reads

$$\dot{\Sigma}_t = \frac{v^2}{\Sigma_t}\sqrt{\alpha^0\left(\sqrt{\pi} + 2\frac{\Sigma_t}{v}\right)} \qquad \alpha^0 = \frac{|g|}{4\sqrt{\pi}\mu\, v^3} , \tag{45}$$

which can be integrated as

$$\sqrt{\sqrt{\pi} + \frac{2\Sigma_t}{\nu}}\left(\frac{\Sigma_t}{\nu} - \sqrt{\pi}\right) = 3\sqrt{\alpha^0}\, t + C^0 \,, \tag{46}$$

where $C^0$ is an integration constant fixed by the initial width of the distribution $\Sigma_0$, $C^0 = \sqrt{\sqrt{\pi} + \frac{2\Sigma_0}{\nu}}\left(\frac{\Sigma_0}{\nu} - \sqrt{\pi}\right)$.

### 4.1.4 Finite-energy solutions

In practice, we know that the energy of the ergodic state computed in section 3 is negative, and therefore we are mainly interested in negative energy solutions. In that case, Eq. (44) with $\tilde{E}_{\text{tot}} < 0$ implies that the width $\Sigma_t$ cannot grow beyond the value $\Sigma^* = \nu\left(\alpha^- + \sqrt{\alpha^-(\alpha^- + \sqrt{\pi})}\right)$ with $\alpha^- = \frac{1}{8\sqrt{\pi}}\frac{|g|}{\nu|\tilde{E}_{\text{tot}}|}$. Furthermore, Eq. (44) can be integrated as

$$F_{\alpha^-}\left(\frac{\Sigma_t}{\nu}\right) = \sqrt{\frac{2|\tilde{E}_{\text{tot}}|}{\mu\nu^2}}\, t + C_- \,, \tag{47}$$

where $F_{\alpha^-}(\xi)$ is defined for $0 \leq \xi \leq \frac{\Sigma^*}{\nu}$ as

$$\begin{aligned}
F_{\alpha^-}(\xi) &= \int^\xi \frac{z}{\sqrt{\sqrt{\pi}\alpha^- + 2\alpha^- z - z^2}}\, dz \\
&= \alpha^- \arcsin\left[\frac{\xi - \alpha^-}{\sqrt{\alpha^-(\alpha^- + \sqrt{\pi})}}\right] - \sqrt{\sqrt{\pi}\alpha^- + 2\alpha^-\xi - \xi^2} \,,
\end{aligned} \tag{48}$$

and $C_- = F_{\alpha^-}(\frac{\Sigma_0}{\nu})$ is an integration constant.

Note that Eq. (47) implies that these solutions exist for finite time intervals since the function $F_{\alpha^-}(\xi)$ has a finite maximal value at $\xi = \frac{\Sigma^*}{\nu}$. For fixed initial conditions, the maximal allowed time for the game scales as $|\tilde{E}_{\text{tot}}|^{-3/2}$ and the solutions converge to the zero energy ones in the limit $\tilde{E}_{\text{tot}} \to 0^-$.

It can be worth noting that in the limit $t \to 0$, the zero energy solution, Eq. (45), the negative energy ones Eq. (47) as well as the positive energy solutions given for completeness in appendix C, Eq. (77), yield a similar behaviour for $\Sigma_t$. This concludes our discussion of the large $\nu$ regime. Next we will address the opposite limit when the healing length is small.

### 4.2 Small $\nu$ regime: Parabolic ansatz

As we have shown in a previous paper [35], in the weak noise, infinite optimization time, limit of the potential-free negative coordination Mean Field Game, the density of players quickly deforms to take the shape of an inverted parabola that scales with time. These parabolic solutions can be interpreted as arising from a low order approximation of a multipolar expansion in a electrostatic representation of the problem [35]. Furthermore, simulations indicate that, under the assumption that the variations of the terminal cost are small compared to $\tilde{u}$, (non scaling) inverted parabolas are still stable solutions of Eqs. (39) with finite optimization time.

Imposing the normalisation condition $\{\int_{-\infty}^{\infty} m(t, x)dx = 1 \ \forall t\}$ we thus consider the ansatz

$$m(t, x) = \begin{cases} \dfrac{3(z(t)^2 - x^2)}{4z(t)^3} & \text{if } z(t) > x \\ 0 & \text{otherwise} \end{cases} \,, \tag{49}$$

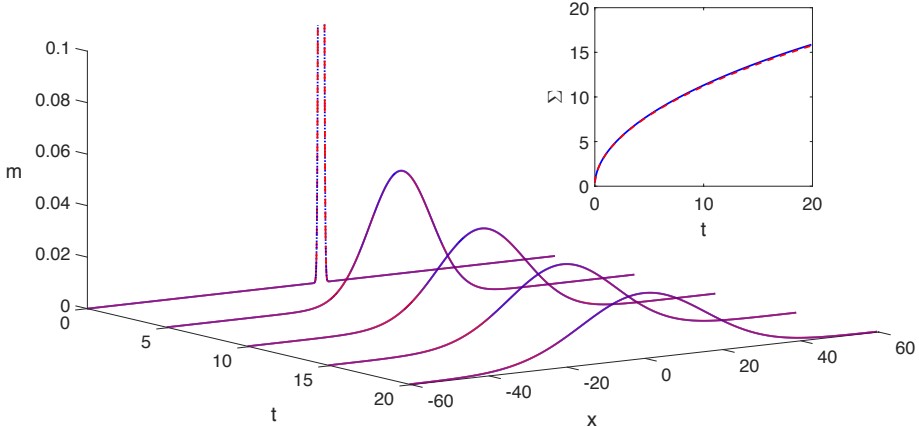

Figure 3: Computational solution of the Gross-Pitaevskii equation (blue dot) and variational Ansatz (red dashed). The inset shows the time evolution of the numerical variance (full) and $\Sigma$ as defined in Eq. (47). In this case $g = -2$, $\sigma = 3.5$, $\mu = 1$ and $T = 20$.

and look for a formal solution outside the singularities in the derivative at $x = \pm z(t)$. It is worth mentioning that such an approach already exists in the realm of cold atoms [45, 46]. However differences arise from the fact that we are dealing with complex time and from the forward-backward structure of Mean Field Games.

In practice, in this subsection, we shall discuss as an "independent problem" an effective potential-free (ie $U_0(x) = 0$) game in the small $\nu$ regime. We furthermore assume that the final condition, at $t = \tilde{T}$, is that of a flat terminal cost $\tilde{c}_{\tilde{T}}(x) = 0$ and that the initial density of agents, at $t = 0$, is essentially a Dirac delta function, i.e. an inverted parabola of the form (49) with $z(t = 0) = z_0 = 0$. Note that, as we will still assume that the healing length $\nu$ is the smallest length size of the problem, this implies that we actually consider here the limit $\nu, z_0 \to 0$ with $z_0 \gg \nu$. In the context of the original game, this effective game will correspond to the expansion phase beyond the healing scale $\nu$. How it will be coupled to the ergodic state or to the small $\nu$ regime will be examined subsequently, but as the conserved energy of the ergodic state is negative, we will consider more specifically this regime.

### 4.2.1 Preliminary definitions

While the Schrödinger representation along with the Gaussian variational ansatz were well-suited to describe a large $\nu$ regime, the hydrodynamic representation is actually more convenient to deal with the small noise limit. In the context of cold atoms, the equivalent of the $o(\sigma^4)$ term in Eqs. (12) is considered to be safely negligible as long as the extension of the condensate is large in front of the healing length $\nu$. Focusing on this weak noise regime (Thomas

Fermi approximation) here amounts to studying the system

$$
\begin{cases}
\partial_t m + \nabla(mv) = 0 \\
\partial_t v + \nabla\left[\dfrac{v^2}{2} + \dfrac{g}{\mu}m + \dfrac{U_0}{\mu}\right] = 0
\end{cases}.
\tag{50}
$$

Going through Madelung substitution shows that we can get away with only neglecting $o(\sigma^4)$ terms while absorbing $o(\sigma^2)$ contributions in the definition of $v$ Eq. (11), which is not as transparent from Eqs. (4).

As we shall see below, we can find exact solutions of Eqs. (50) assuming the parabolic form (49), and, therefore, we shall not need to resort to the action (13) to derive the corresponding dynamics.

### 4.2.2 Elementary integration of the hydrodynamic representation

In the $U_0(x) = 0$ limit the expression of the velocity associated to a parabolic distribution Eq. (49) can easily be extracted from the continuity equation in (50). Integrating over $[-\infty; x]$ and taking into account that $m$ vanishes at infinity, we get

$$
v(t,x) = \frac{z'(t)}{z(t)}x \,.
\tag{51}
$$

To derive the time evolution of $z(t)$, we insert the explicit forms of $m(t,x)$ and $v(t,x)$ in the second equation of Eqs. (50), yielding

$$
z''(t) = \frac{3g}{2\mu z(t)^2} \,.
\tag{52}
$$

This closely resembles what can be found when dealing with expanding Bose Einstein condensates (BEC) [46], one main difference lying in the fact that the multiplicative constant in front of $1/z^2$ is negative in the context of Mean Field Games but positive in the context of Bose Einstein condensates.

Eq. (52) can be integrated as

$$
z'(t)^2 = -\frac{3g}{\mu}\left[\frac{1}{z(t)} + \frac{\epsilon}{z_*}\right].
\tag{53}
$$

For commodity the integration constant has been written as $3|g|\epsilon/\mu z_*$ and $\epsilon$ can take the value $-1$, $0$ or $1$. We shall see below the values $-1$, $0$ or $1$ of $\epsilon$ correspond to negative, $0$ or positive energies, and that in the $\epsilon = -1$ (negative energy) case, $z_*(> 0)$ can be interpreted as $z(\tilde{T})$ for the effective game. In the BEC context, only the positive $\epsilon$ case is relevant [46], and the fact that, here, zero or negative $\epsilon$ have to be considered as well, which allows for new sets of solutions, constitutes another important difference.

### 4.2.3 Characterisation of $z(t)$

To solve this equation, let us introduce two functions $\xi^+(y) > 0$ and $\xi^-(y) \in [0;1]$, associated with $+1$ and $-1$ values of $\epsilon$, implicitly defined through the relations

$$
\sqrt{\xi^+(y)(1 + \xi^+(y))} - \operatorname{argsinh}\sqrt{\xi^+(y)} = y \quad \forall y > 0 \,,
\tag{54}
$$

and

$$
\arcsin\sqrt{\xi^-(y)} - \sqrt{\xi^-(y)(1 - \xi^-(y))} = y \quad \forall y \in [0, \tfrac{\pi}{2}] \,.
\tag{55}
$$

We also define a third function $\xi^0(t)$ given explicitly as

$$\xi^0(y) = \left(\frac{3y}{2}\right)^{2/3} \quad \forall y > 0 \,, \tag{56}$$

which corresponds to the $\epsilon = 0$ solution discussed in [35]. It is worth noting that all three functions are monotonous increasing functions of time and have the following properties

$$\begin{cases} \xi^+(0) = \xi^-(0) = \xi^0(0) = 0 \\ \xi^+(y) > \xi^0(y) \quad \forall y \\ \xi^0(y) > \xi^-(y) \quad \forall y \in \,]0, \frac{\pi}{2}] \\ \xi^+(y) \approx \xi^0(y) \approx \xi^-(y) \quad \text{as } y \to 0 \end{cases} .$$

We can now write the different solutions of Eq. (53) in terms of the above functions. Even if we only consider repulsive interactions, because of the square power in Eq. (53), its solutions can either be increasing or decreasing. There are three families of increasing solutions

$$z(t) = \begin{cases} z_* \xi^+(\alpha z_*^{-3/2} t) & \text{if } \epsilon = 1 \\ \xi^0(\alpha t) & \text{if } \epsilon = 0 \\ z_* \xi^-(\alpha z_*^{-3/2} t) & \text{if } \epsilon = -1 \end{cases} , \tag{57}$$

where $\alpha = \sqrt{-3g/\mu}$. The reciprocal three families of decreasing solutions are irrelevant to our discussion as they will not ultimately lead to the ergodic state introduced section 3. We still provide a succinct analysis of those solutions in appendix D for the sake of completeness.

Let us address how the boundary conditions of our effective game constrain the solution within the family (57). The aforementioned initial condition that the density of agents starts as a Dirac delta function imposes that $z(t=0) = 0$ is already implemented in Eq. (57). Consider now the the terminal boundary condition, i.e. the fact that at $\tilde{T}$ the terminal cost is flat. Recalling that $v = -\nabla u/\mu + o(\sigma^2)$, the expression of the velocity (51), implies that the terminal cost $c_{\tilde{T}}(x) = u(\tilde{T}, x)$ can be constant only if the time derivative of $z(t)$ is zero. According to Eq. (53), this is only possible if $\epsilon = -1$ and $z(t) = z_*$. Hence, the study of the effective game we consider here can be reduced to that of "-" type solutions and we deduce that $z_* = z(\tilde{T})$. Now, one can check easily from Eq. (55) that $\xi^-(\pi/2) = 1$ (which is compatible with the fact that $\xi^-(y) \in [0; 1]$ is an increasing monotonous function defined for $y \in [0, \pi/2]$). From Eq. (57) we infer

$$z(\tilde{T}) = z_* \quad \Rightarrow \quad \alpha z_*^{-3/2}(\tilde{T}) = \frac{\pi}{2} \,. \tag{58}$$

This yields a relation between the final time of the effective game $\tilde{T}$ and the final extension of the distribution of players

$$\tilde{T} = \frac{\pi z_*^{3/2}}{2\alpha} \,. \tag{59}$$

The duration of the effective game, i.e. the time it takes to go from a narrow, delta-like initial density of agents, to a flat terminal cost, thus determines the parameter $z_*$, and therefore fixes which member of the family Eq. (57) has to be considered.

Inserting Eq. (57) in the ansatz (49) and (51), directly yields explicit expressions for $m$ ans $v$, which, as illustrated in Figure (4) provide satisfactory approximations, even though the noise $\sigma$, and thus the healing length $v$, is not strictly zero (see captions for details).

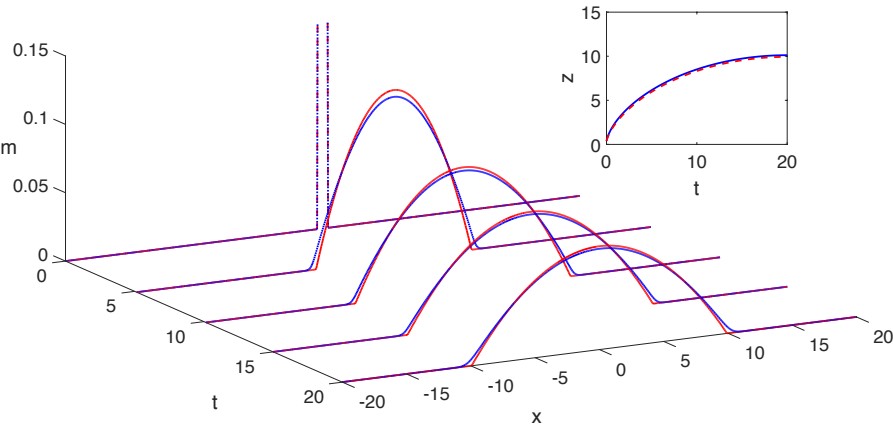

Figure 4: Computational solution of the Gross-Pitaevskii equation (dot) and parabolic ansatz (dashed). The inset shows the time evolution of z numerically (full) and analytically (dashed). In this case, we have chosen $g = -2$, $\sigma = 0.45$ and $\mu = 1$, meaning $\nu \approx 0.02$. The actual (numerical) game takes place from $t = 0$, when it starts as an inverted parabola of extension 0.4, to $t = T = 20$ when the terminal cost is flat. The effective game starts at time $t \approx -0.07$ as a Dirac delta function and its effective duration is $\tilde{T} \approx 20.07$. The only difference between the numerical results and the parabolic ansatz comes from the fact that $\sigma$ is non-zero in the simulation. This figure also illustrates how the Thomas-Fermi approximation becomes more and more effective as the typical extension of the density becomes larger in front of $\nu$.

### 4.2.4 Energy of the system

The energy plays a crucial role in the dynamics of the spreading of the players and its conservation will be the key property we will use to match the different regimes of approximation. Because we ultimately want to link this regime to the ergodic state described in section 3, we will focus on negative energy only. In the potential free regime, the energy contains two terms, one is the "kinetic energy" (associated with the diffusion term), the other comes from the interactions. Dropping the $o(\sigma^4)$ term in the definition Eq. (15) of the kinetic energy, we thus have $E = E_{\text{kin}} + E_{\text{int}}$, with

$$
\begin{cases}
E_{\text{kin}} = \dfrac{\mu}{2} \displaystyle\int_{-z}^{z} m v^2 dx \\[2mm]
E_{\text{int}} = \displaystyle\int_{-z}^{z} \dfrac{g m^2}{2} dx
\end{cases} \tag{60}
$$

As the energy is conserved, it can be evaluated at any time, and particularly at the end of the effective game. If $\epsilon = 0$, $z \to \infty$ as $t \to \infty$ and it becomes clear that, in this case, $E = 0$. A similar reasoning would show that, if $\epsilon = +1$, $E \sim 1/z_*^2 > 0$. When $\epsilon = -1$, however, we can evaluate the energy at $t = \tilde{T}$, when $z = z_*$ and $v = 0$, which trivially implies that, at that point

and within the Thomas-Fermi approximation, the kinetic energy is zero. Inserting Eq. (49) with $z(t) = z_*$ into the second equation of (60) we get

$$
\begin{cases}
E_{\text{kin}}^-(\tilde{T}) = 0 + o(\sigma^4) \\
E_{\text{int}}(\tilde{T}) = \dfrac{3g}{10z_*}
\end{cases}
,
\tag{61}
$$

which, using Eq. (59) implies

$$
E = \frac{3g}{10z_*} = \frac{3g}{10}\left(\frac{2\alpha\tilde{T}}{\pi}\right)^{-2/3}.
\tag{62}
$$

For the effective game we consider here – narrow initial density, flat terminal cost $v(\tilde{T}) = 0$, small $\nu$ regime, individual gain $U_0(x) = 0$ – there is a strong link between the duration of the game $\tilde{T}$ and the energy $E$. In some sense $\tilde{T}$ monitors the dynamics of the spreading of the players completely, and takes the same role as $\tilde{E}_{\text{tot}}$ did in the large $\nu$ regime. As such, finite games with flat terminal cost correspond to non-0 energy and there is a one-to-one relation between $\tilde{T}$ and $E$.

This finishes our analysis of the small $\nu$ regime, and more generally of the expansion regime. The next section will now address ways to relate those transient times to the ergodic state.

# 5  The full game

As stressed at the beginning of section 3, the existence of an ergodic state in the long optimization time limit makes it possible to effectively split the full optimization problem into two decoupled ones, the first linking the initial condition to the ergodic state, and the second the ergodic state to the final boundary condition. Here, as the second can be analyzed following essentially the same lines, we consider only the first of these transient regime. In this section, we thus examine how the regimes of approximation discussed in the two previous section couple with one another.

We start in section 5.1 by first addressing, once again, an effective game, in the vein of the one we studied in section 4.2, but assuming a finite value of healing length so that players are initially distributed on a distance much smaller than $\nu$. This will allow us to focus on the transition from a large to a small $\nu$ regime during the initial stages of the game. Then, in section 5.2 we will consider the the transition from this initial phase of expansion towards an ergodic state.

## 5.1  Matching small and large $\nu$ regimes

As mentioned above, we consider here, just as in section 4.2, an effective potential-free game of duration $\tilde{T}_V$, with flat terminal cost and an initial distribution of agents which width $\Sigma_0$ is much smaller that the healing length $\nu$. We furthermore assume that the optimization time is large enough so that, at the end of the game, the density of player has spread on a distance much larger than $\nu$.

Under those assumptions, we can distinguish two main phases the effective game will go through: an initial phase which can be described by the Gaussian ansatz introduced section 4.1 and, at the end of the game, a terminal phase for which the density of agents will follow the parabolic ansatz of section 4.2. Between those two phases, the density will transition from a Gaussian-like distribution to an inverted parabola. The precise shape of the density during the

crossover is complicated to describe, and will not be addressed here, but we shall see that we can still describe the dynamics of the spreading of the players across the two regimes.

To proceed, let us introduce a couple of quantities that will characterise the dynamics. The first one is the total energy $E$ of the system, a conserved quantity, which is common to both regimes. The second is the time $t_{\text{tr}}$ at which the system will transition from the Gaussian regime to the parabolic one.

Seen from within the initial, Gaussian, description, the transition time $t_{\text{tr}}^{\text{G}}$ can be defined by the condition

$$\Sigma(t_{\text{tr}}^{\text{G}}) = \nu, \tag{63}$$

which through Eq. (47) provides a relation between $E$ and $t_{\text{tr}}^{\text{G}}$

$$F(8E, -2g/\sqrt{\pi}, \mu\sigma^4; \nu) - F(8E, -2g/\sqrt{\pi}, \mu\sigma^4; \Sigma_0) = \frac{t_{\text{tr}}^{\text{G}}}{2\sqrt{\mu}}. \tag{64}$$

In the parabolic description, the duration $\tilde{T}_{IV}$ of the effective game of section 4.2 can be inferred from the expression for the energy, Eq. (62)

$$\tilde{T}_{IV} = \frac{\pi}{2\alpha}\left(\frac{3g}{10E}\right)^{3/2}. \tag{65}$$

On this side of the transition, the transition time $t_{\text{tr}}^{\text{para}}$ is thus obtained by the condition

$$\frac{z(t_{\text{tr}}^{\text{para}} - t_0^{\text{para}})}{\sqrt{5}} = \nu, \tag{66}$$

where $z/\sqrt{5}$ is the standard deviation of the parabolic distribution Eq. (49), and $t_0^{\text{para}} = \tilde{T}_V - \tilde{T}_{IV}$ the fictitious time at which the parabolic evolution appears to have started (from an initial Dirac delta shape) seen from the large $z$ side of the transition. From Eq. (57) this implies that $\sqrt{5}\nu/z_* = \xi^-(\alpha z_*^{-3/2}(t_{\text{tr}}^{\text{para}} - t_0^{\text{para}}))$. Inserting this into Eq. (55), we obtain now a relation between $t_{\text{tr}}^{\text{para}}$ and $z_*$

$$\frac{\alpha}{z_*^{3/2}}(t_{\text{tr}}^{\text{para}} - t_0^{\text{para}}) = \arcsin\sqrt{\frac{\sqrt{5}\nu}{z_*}} - \sqrt{\frac{\sqrt{5}\nu}{z_*}\left(1 - \frac{\sqrt{5}\nu}{z_*}\right)}, \tag{67}$$

which, given the fact that $z_*$ and $E$ are linked through Eq. (62) is actually a relation between $t_{\text{tr}}^{\text{para}}$ and $E$.

The self-consistent condition $t_{\text{tr}}^{\text{para}} = t_{\text{tr}}^{\text{G}}$ then implies that Eqs. (64)-(67) fix both the energy $E$ and the transition time $t_{\text{tr}}$, and thus solve the game we are considering in this subsection.

Knowing the energy, as illustrated in Figure (5), one can reconstruct the evolution of the variance of the Gaussian distribution at small times using Eq.(47) and, then, of the width of the inverted parabola using Eq. (57). Figure (6) gives further indication that both the Gaussian and parabolic ansatz yield good result to evaluate not only the spreading of the players but also the shape of the distribution in this configuration. The two regimes overlap when $\Sigma_t$ is of order $\nu$ and either approximation regime gives a fairly accurate description of the phenomenon. However, near the end of the game both approximations become less and less accurate due to the vicinity of the terminal condition, which, because $\sigma$ is small but positive, is not identically zero, $\nu_T(x) = 0 + o(\sigma^2)$.

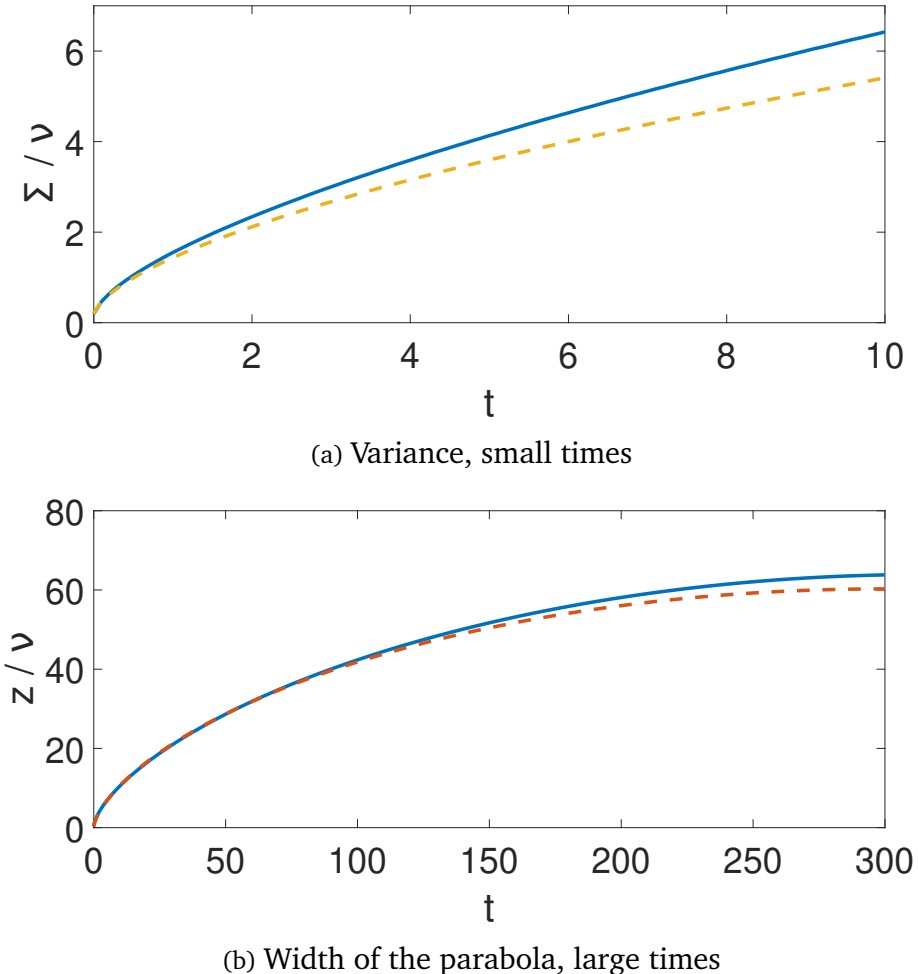

(a) Variance, small times

(b) Width of the parabola, large times

Figure 5: Time evolution of the variance (a) and the width of the parabola (b). The numerical solution for the density of players has been numerically fitted with a Gaussian and an inverted parabola, full curves are obtained through the extraction of the fitting parameters. Dashed curves are obtained using either the Gaussian or parabolic ansatz with energy $E = -9.95 \times 10^{-3}$ computed through the self-consistent condition. Parameters for this figure are $g = -2$, $\sigma = 1.2$, $\mu = 1$, $\nu = 1$, $\Sigma_0 = 0.2$ and $T = 300$. One can check that the Gaussian ansatz produces satisfactory results for small times, up to $\Sigma \approx 2\nu$, while the parabolic ansatz yields good results for large $z$.

## 5.2 Matching transient and ergodic states

We now turn back to the complete game of Eqs. (4), or more specifically the first half of that game linking the initial distribution of agents to the ergodic state. We specialize moreover to the case of a narrow initial condition, of width $\Sigma_0 \ll \nu$, for the distribution of agents. It should also be noted that we will assume that the maximum of the external gain $U_0$ coincides with the center of mass of the initial distribution, so that we do not have to take its motion into account. The system will, therefore, initially go through an expansion phase, during which we will neglect the individual gain / potential $U_0(x)$, and will successively traverse the large $\nu$ and the small $\nu$ regimes before reaching the ergodic state. Our goal here is to understand how to connect those.

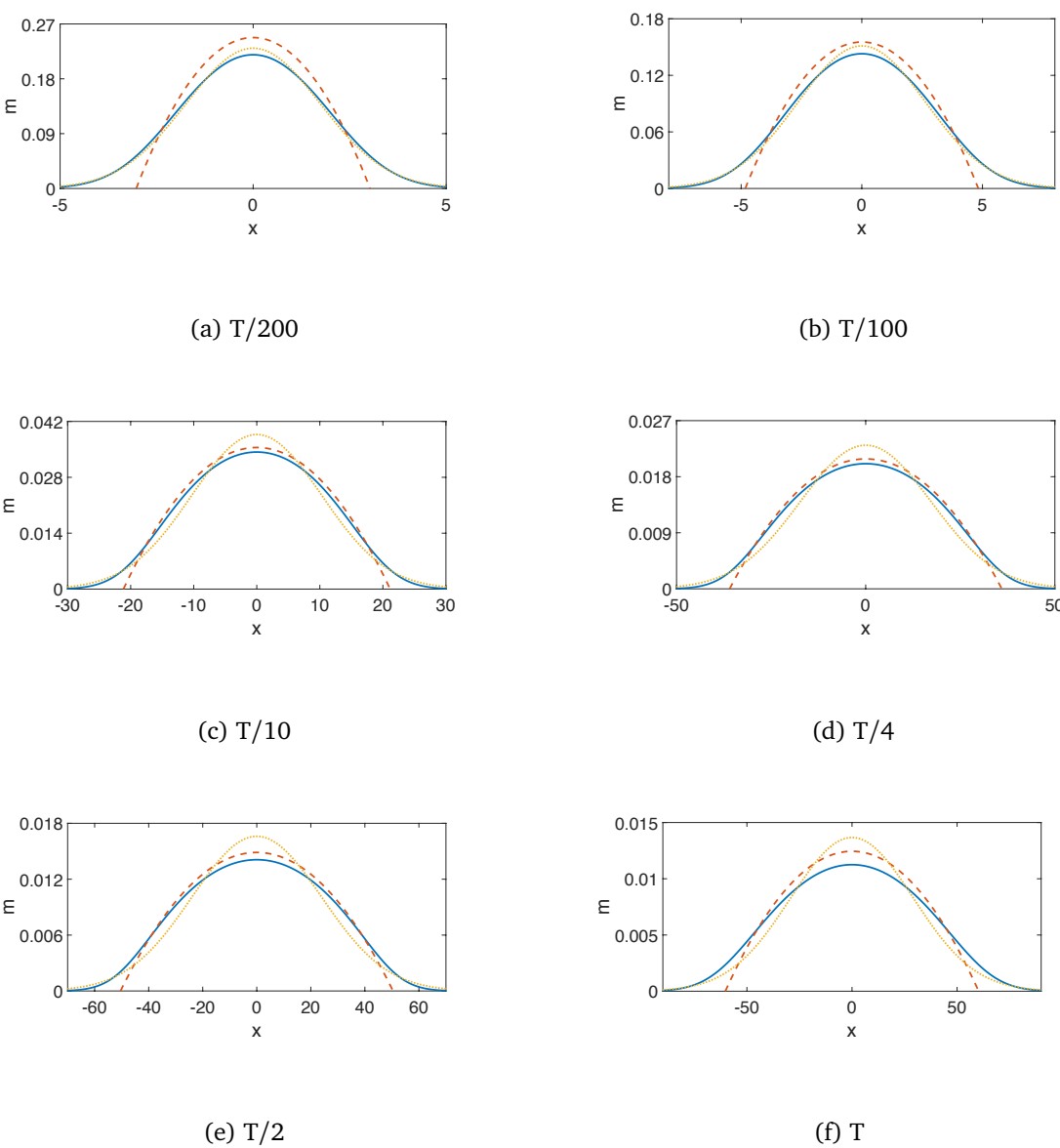

(a) T/200

(b) T/100

(c) T/10

(d) T/4

(e) T/2

(f) T

Figure 6: Density of players at different times, numerical results are plotted (solid line) along with the Gaussian (dotted line) and the parabolic ansatz (dashed line). At the beginning of the game, Figs (6a) and (6b), the Gaussian ansatz is the most accurate. Then in the middle of the game, Figs (6c) and (6d), the parabolic constitutes a better approximation. At the end of the game, Figs (6e) and (6f), the parabolic ansatz becomes less and less accurate as we near the terminal condition. Here $g = -2$, $\sigma = 1.2$, $\mu = 1$, $\nu = 1$, $\Sigma_0 = 0.2$ and $T = 300$, while $E = -9.95 \times 10^{-3}$ has been computed through the self-consistent condition.

In this configuration, the energy $E$ is completely fixed by the ergodic state

$$E = E_{\text{er}} = \frac{g}{2} \int_{\mathbb{R}} m_{\text{er}}^2 dx + \int_{\mathbb{R}} m_{\text{er}} U_0 dx \ . \tag{68}$$

The initial "large $\nu$" expansion phase is therefore completely fixed by $E$ and $\Sigma_0$ through Eq. (47), which in turn fixes the transition time $t_{\text{tr}}$ between the large and the small $\nu$ regimes

through Eq. (64).

Once in the large $\nu$ regime, the energy $E$ again fixes the duration $\tilde{T}_{IV}$ of the effective game of section 4.2. The only parameter that remains to be fixed is the effective beginning time $t_0^{\text{para}}$ of that effective game which is given by Eq. (67) (with, according to Eq. (62), $z_* = 3g/10E$).

Naturally, because one has to take the external gain into account when nearing the ergodic state, the final extension of the effective game $z_*$ does not correspond to the extension the ergodic state $z_{\text{er}}$ and its duration $\tilde{T}_{IV}$ does not correspond to typical duration $\tau_{\text{er}}$ of the transient time leading to the ergodic state. However, those respective quantities are of same order as long as, in the ergodic state, interaction energy and potential energy are comparable. No matter the external gain, as mentioned in section 3.2

$$E_{\text{int}}^{\text{er}} \sim \frac{g}{z_{\text{er}}} \; . \tag{69}$$

Hence, if interaction energy represents a set proportion $p$ of the total energy, $E_{\text{int}}^{\text{er}} = pE_{\text{er}}$, $z_{\text{er}}$ should be of order $z_*/p$. And, noting that $\tilde{T}_{IV} \sim z_*^{3/2}$, we can infer that $\tau_{\text{er}}$ should not be too far-off from $\tilde{T}_{IV}/p^{3/2}$. In the particular case of a quadratic external gain $U_0(x) = -\mu\omega_0^2 x^2/2$, we can easily compute the ratio between $E_{\text{int}}^{\text{er}}$ and $E_{\text{pot}}^{\text{er}}$

$$\frac{E_{\text{int}}^{\text{er}}}{E_{\text{pot}}^{\text{er}}} = 2 \quad \Rightarrow \quad E_{\text{int}}^{\text{er}} = \frac{2}{3}E \; , \tag{70}$$

result which is completely independent of the values of $g$, $\mu$ or $\omega_0$. The ergodic density is then an inverted parabola of width $z_{\text{er}} = 3z_*/2$ and $\tau_{\text{er}}$ is of order $\tilde{T}_{IV}(3/2)^{3/2}$. This is illustrated Fig. (7).

What the effective game provides, in this context, is not a quantitatively precise description but a good qualitative estimation of what actually happens during the beginning of the game.

# 6 Conclusion

Mean Field Games constitute a challenge because of their unusual forward-backward structure. In this paper we presented a simple, heuristic, yet efficient method to describe negatively coordinated Mean Fields Games in one dimension, leaning heavily on the notion of ergodic state introduced by Cardaliaguet [34]. The existence of this ergodic state proves to be of paramount importance as it allows the initial and final conditions to essentially decouple. The problem of finding a way to link initial and final conditions, both arbitrary, simplifies as it becomes a problem of finding a way to link either to a generic ergodic state. Making first use of the mapping to the non-linear Schrödinger equation as introduced in [32], and then of the hydrodynamic representation from [35], we were able to identify different regimes of approximation and put forward adequate ansätze to reconstruct the whole game. Results from those ansätze have been compared to numerical solutions, for parameters in their domain of application, and are highly satisfactory as well as easily computed.

# A  Derivation of the semi-classical approximation for quadratic external potential

Deriving a semi-classical approximation for the (linear) Schrödinger equation amounts to solving Eq. (27) up to second order in $\sigma$, assuming a solution of the form $\Psi_{\text{SC}}(x) = \psi(x)\exp\left(\frac{S(x)}{\sqrt{\mu\sigma^4}}\right)$.

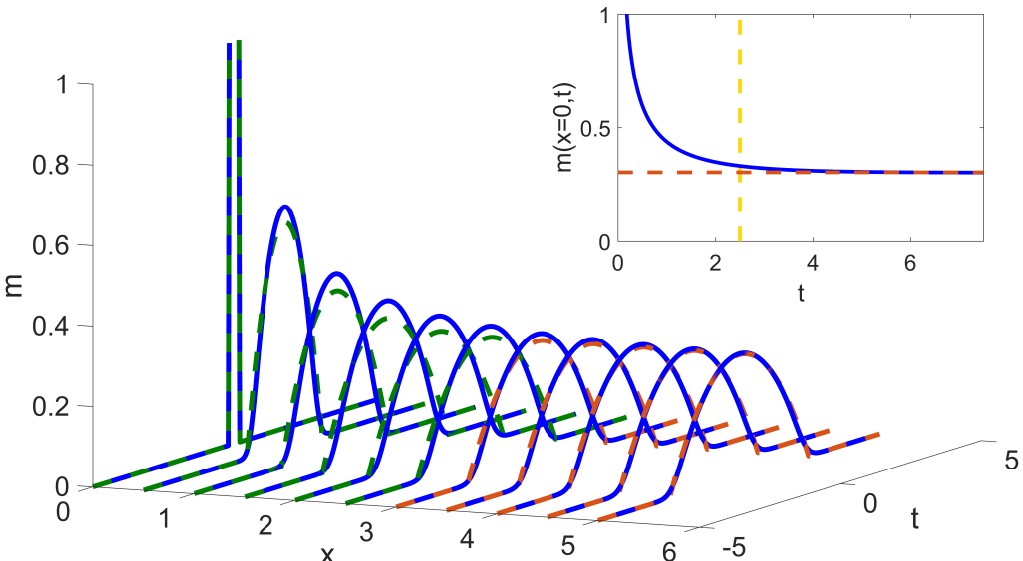

Figure 7: The full blue line represents the numerical density of players, the dashed green line is obtained through a parabolic ansatz of intrinsic time $\overline{\tau} = \tilde{T}_{IV}(3/2)^{3/2} = 2.5$ and the dashed red line corresponds to the ergodic density. In the inset the full line shows the time evolution of the maximum of the player density $m(x = 0, t)$, while the dashed horizontal line is set at $m_{\mathrm{er}}(0)$, maximum of the density during the ergodic state, and the dotted vertical at $t = \overline{\tau} = 2.5$. Here $g = -2$, $\sigma = 0.4$, $\mu = 1$, $\omega_0^2 = 0.2$, $E = -0.36$ and $T = 15$.

**Order $\sigma^0$**

At zeroth order Eq. (27) reduces to the Hamilton-Jacobi equation

$$\frac{(\partial_x S)^2}{2} + (U_0(x) + \lambda) = 0 \, , \tag{71}$$

which, for the kind of one dimensional problem we consider here, can be reduced to a simple quadrature. Taking once again the example of a quadratic potential $U_0(x) = -\mu\omega_0^2 x^2/2$, with a turning point located at $X = \sqrt{2\lambda/\mu\omega_0^2}$, we get

$$
\begin{aligned}
S(x) &= \int_X^x \sqrt{2(-U_0(s) - \lambda)} ds \\
&= \frac{\lambda}{\sqrt{\mu\omega_0^2}} \left[ x\sqrt{\frac{\mu\omega_0^2}{2\lambda}} \sqrt{x^2 \frac{\mu\omega_0^2}{2\lambda} - 1} - \mathrm{argcosh}\left( x\sqrt{\frac{\mu\omega_0^2}{2\lambda}} \right) \right] .
\end{aligned}
\tag{72}
$$

We note that the $-(U_0 + \lambda)$ term under the square root is positive on the right of the turning point, and thus for the whole range of validity of Eq. (28), $x \gg X$.

In the case of a Langer-type uniform approximation, however, one has to specify how to analytically continue the square root for negative value of $(U_0 + \lambda)$ on the left side of the turning point. We therefore introduce the notations

$$S_{\mathrm{right}}(x) = S(x) \, , \tag{73}$$

valid for $x > X$ and

$$
\begin{aligned}
S_{\text{left}}(x) &= \int_x^X \sqrt{2(\lambda + U_0(s))} ds \\
&= \frac{\lambda}{\sqrt{\mu \omega_0^2}} \left[ \frac{\pi}{2} - x \sqrt{\frac{\mu \omega_0^2}{2\lambda}} \sqrt{1 - x^2 \frac{\mu \omega_0^2}{2\lambda}} - \arcsin\left( x \sqrt{\frac{\mu \omega_0^2}{2\lambda}} \right) \right],
\end{aligned}
\tag{74}
$$

valid for $x < X$. We will not provide the details of the computations for the uniform approximation, rather referring the reader to Langer's seminal paper [42]. The result of this uniform approximation is expressed in terms of $S_{\text{right}}$ and $S_{\text{left}}$ as Eq (29).

**Order $\sigma^2$**

At first order in $\sigma^2$, Eq. (27) becomes

$$
\partial_{xx} S(x) \psi(x) + 2 \partial_x S(x) \partial_x \psi(x) = 0,
\tag{75}
$$

which is solved as

$$
\psi(x) = \frac{C^{1/4}}{\sqrt{\partial_x S(x)}} = \left[ \frac{C}{2(-U_0(x) - \lambda)} \right]^{1/4},
\tag{76}
$$

where the last equality derives from Eq. (71), and $C$ is a constant that has to be obtained numerically.

# B Proof that the operator $\hat{D}$ has only real non-negative eigenvalues

In this appendix, we prove that the operator $\hat{D}$ introduced in Eq. (35) has only real non-negative eigenvalues.

Consider any two function with compact support $(\varphi, \varphi')$. Integrating by part twice gives that $\langle \varphi | \hat{D} | \varphi' \rangle = \int d\mathbf{x} \varphi(\mathbf{x}) \hat{D}[\varphi'(\mathbf{x})] = \int d\mathbf{x} \hat{D}[\varphi(\mathbf{x})][\varphi'(\mathbf{x})] = \langle \varphi' | \hat{D} | \varphi \rangle$. $\hat{D}$ is therefore a real symmetric operator, and has only real eigenvalues.

Furthermore, introducing $\epsilon_i$ eigenvalue of $\hat{D}$, and $\varphi_i(\mathbf{x})$ the corresponding eigenvector, we have $\langle \varphi_i | \hat{D} | \varphi_i \rangle = \epsilon_i \int d\mathbf{x} \varphi_i^2(\mathbf{x}) = \int d\mathbf{x} [\nabla \varphi(\mathbf{x})]^2 m_{\text{er}}(\mathbf{x})$. Since $\varphi(\mathbf{x})^2$, $[\nabla \varphi(\mathbf{x})]^2$, and $m_{\text{er}}(\mathbf{x})$ are all positive quantities, this implies that $\epsilon_i$, too, has to be positive.

# C Positive energy solutions

For completeness, we also provide solution of Eq. (44) in the case of positive energy, $E_{\text{tot}} > 0$. Defining $\alpha^+ = \frac{1}{8\sqrt{\pi}} \frac{|g|}{\nu E_{\text{tot}}}$, the time dependent width $\Sigma_t$, solution of Eq. (44) can be written implicitly as

$$
G_{\alpha^+}\left( \frac{\Sigma_t}{\nu} \right) = \sqrt{\frac{2E_{\text{tot}}}{\mu \nu^2}} \, t + C_+,
\tag{77}
$$

where $G_{\alpha^+}(\xi)$ is defined as

$$
G_{\alpha^+}(\xi) = \begin{cases} \sqrt{\xi^2 + 2\alpha^+ \xi + \sqrt{\pi}\alpha^+} - \alpha^+ \text{argsinh}\left[ \frac{\xi + \alpha^+}{\sqrt{\alpha^+(\sqrt{\pi} - \alpha^+)}} \right] & \text{if } \alpha^+ < \sqrt{\pi} \\ \sqrt{\xi^2 + 2\alpha^+ \xi + \sqrt{\pi}\alpha^+} - \alpha^+ \text{argcosh}\left[ \frac{\xi + \alpha^+}{\sqrt{\alpha^+(\alpha^+ - \sqrt{\pi})}} \right] & \text{if } \alpha^+ > \sqrt{\pi} \end{cases},
\tag{78}
$$

and $C_+ = G_{\alpha^+}(\frac{\Sigma_0}{\nu})$ is the integration constant.

## D   Decreasing solutions of the effective game

As mentioned in section 4.2 we provide here expressions for the decreasing families of solutions of the effective game

$$z(t) = \begin{cases} z_* \xi^+(\alpha z_*^{-3/2}(t_0-t)) & \text{if } \epsilon = 1 \\ \xi^0(\alpha(t_0-t)) & \text{if } \epsilon = 0 \\ z_* \xi^-(\alpha z_*^{-3/2}(t_0-t)) & \text{if } \epsilon = -1 \end{cases} . \tag{79}$$

Contrary to increasing solutions, decreasing solutions can only be defined on $[0, t_0]$, and with $t_0 < \frac{\pi z_*^{3/2}}{2\alpha}$ if $\epsilon = -1$. Using those properties we can also construct a mixed type solution by patching together an increasing "-" type solution with a decreasing one of same $z_*$

$$z(t) = \begin{cases} z_* \xi^-(\frac{\pi}{2} + \alpha z_*^{-3/2}(t-T_m)) & \text{for } 0 \le t \le T_m \\ z_* \xi^-(\frac{\pi}{2} - \alpha z_*^{-3/2}(t-T_m)) & \text{for } T_m \le t \le T \end{cases} , \tag{80}$$

with $T_m$ the the time at which the solutions starts decreasing, with $T - \frac{\pi z_*^{3/2}}{2\alpha} \le 0 \le T_m \le \frac{\pi z_*^{3/2}}{2\alpha}$.

Increasing "+", decreasing or mixed type solutions can all be observed numerically. They refer to configurations where variations of the terminal cost are important in front of $\tilde{u} = \mu\sigma^2$ and can be used to describe the end of the game, just like increasing "-" solutions can be used as approximations of its beginning. For these reasons they fall outside the scope of this article, still we mention them, once again, for the sake of completeness.

## Acknowledgments

This work has benefited from discussions with Max-Olivier Hongler, Olivier Guéant, and Filippo Santambrogio. We are especially thankful to Nicolas Pavloff who has introduced us to many aspects of the physics of the non-linear Schrödinger equation that were relevant to this work.

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
