# Peer review of "Schrödinger approach to Mean Field Games with negative coordination"

_SciPost Physics, doi:SciPost Phys. 9, 059 (2020)_

## Round 1 · Referee Report · Anonymous (Referee 1) · 2020-7-13

# Report on the manuscript entitled Schrödinger approach to Mean Field Games with negative coordination

Th. Bonnemain, D. Ullmo and Th. Gobron

As it has been formerly shown by the authors, the Schrödinger dynamics (SD) and the meanfield games (MFG) evolutions formally exhibit close similarities. In the present paper, the authors take full advantage of such similarities to construct analytically approximative solutions of a full class of quadratic MFG with negative MF coordination, linear in the players density $m(\mathbf{x})$ with and/or without external potential $U_0(\mathbf{x})$. Since MFG's involve nonlinear pde's and a forward-backward structure in time (i.e. specified by an initial condition for the players density and a final condition for the value functions), analytic solutions are generally very difficult to construct. The present work offers a new light on analytical possibilities of solving these MFG's and this for negative MF coordination which is yet unexplored. The content of this contribution is original and it offers some intuitive understanding on how the MFG evolves for different time regimes. In particular, the influence of the initial and final conditions on the transient development of the MFG's behaviour is clearly explained. The paper is mostly technical and its presentation offers all necessary calculation details to attract the attention of a theoretical physicist readership. Some new results like the particularly simple parabolic players density with time-dependent support expressed in Eq.(49) are definitely interesting.

**I recommend to accept this study for publication for a theoretical physicist readership**.

## Notation details and misprints

Could the authors consider the following list details:

1) Notation issue. It is not clear always clear to me when the development is valid for players evolving in $\mathbb{R}^d$ for $d > 1$ or the evolution is

restricted to scalar situations $d = 1$. For example in section 3, Eq.(31) is written for $d = 1$ whereas Eq.(35) which follows seems to hold for $\geq 1$ (and so is also the case for Eqs.(32) and (33)). In section 4 , one is limited to $d = 1$, etc...

2) **Misprint** ? In Eq.(8) on page 5. Does the term $(U_0 + g\Gamma\phi)\Phi$ should read $(U_0 + g\Gamma\Phi)\Phi$ ?

3) In section 3.2. [...] *by looking at the expression for the energy (15), one can note that a natural length scale appears* [....]. Could the authors give a hint to see how one spontaneously sees this $L$.

4) In Eq.(26), Eq. $m_{er} = \frac{\lambda + U_0}{|g|}$ would it be perhaps be more clear if one writes $m_{er}(\mathbf{x}) = \frac{\lambda + U_0(\mathbf{x})}{|g|}$ (or perhaps $m_{er}(x) = \frac{\lambda + U_0(x)}{|g|}$)?

5) **Misprint** ? Between Eqs.(28) and (29). Does $X = \sqrt{\frac{2\lambda}{\mu\omega}}$ should perhaps read as $X = \sqrt{\frac{2\lambda}{\mu\omega_0}}$ ?

6) After Eq.(32). The perturbations are infinitesimal fields so one means that $\delta m = \delta m(\mathbf{x}, t)$ and $\delta v = \delta v(\mathbf{x}, t)$ and $\delta m_0$ stands for $\delta m_0(\mathbf{x})$ and hence:

$$\partial_t \delta m(\mathbf{x}, t) = -\nabla \left[ m_{er}(\mathbf{x}) \delta v(\mathbf{x}, t) \right],$$
$$\partial_t \delta v(\mathbf{x}, t) = -\frac{g}{\mu} \nabla \delta m(\mathbf{x}, t)$$

and then the same remark as expressed in point 1) holds again. Indeed in Eq.(37) and (38) one has $\delta m(x, t) := \delta m(x) e^{\pm\omega t}$ and so $\mathbf{x} \mapsto x$ in these expressions.

---

## Round 1 · Referee Report · Anonymous (Referee 2) · 2020-7-19

Strengths

1) Well performed and interesting, on a topical subject
2) Well written, review like paper on a trans-disciplinary subject

Weaknesses

1) Method seems restricted to the very special case of quadratic games
2) Quite technical paper

Report

I suggest acceptance of this paper essentially as it

Requested changes

Maybe discuss if and how these methods can be adapted to more general cases: non quadratic games; time dependent potential U_0.

---

## Round 2 · Author Response

please find enclosed a revised version of our manuscript.
We thanks both referees for their very positive appreciation of our work, and for their detailed reading of our manuscript. Referee 1 in particular made a list of rather precise suggestions of changes, that we have integrally implemented. We detail the list of change below.
Best regards
Denis Ullmo (for the authors)

---

## Round 2 · List of Changes

ii) We have clarified the sentence in section 3.2 to make clear that the scale that emerges is indeed the healing length $\nu$.
ii) We have corrected misprints in Eq. (8) and in the expression of X between Eqs.(28) and (29), and have clarified the notations in Eqs. (26) and (32).

---

## Editorial Decision

published